# Myriocin Restores Metabolic Homeostasis in dAGE-Exposed Mice via AMPK-PGC1α-Mediated Mitochondrial Activation and Systemic Lipid/Glucose Regulation

**DOI:** 10.3390/nu17091549

**Published:** 2025-04-30

**Authors:** Libo He, Jinye Dang, Jingjing Li, Hairui Xue, Jiaxiu Cai, Guohua Cheng, Yuhui Yang, Zhiyi Liu, Binghua Liu, Yali Dai, Yu Zhang, Yating Huang, Yiran Sun, Jinlin Guo, Ke Liu

**Affiliations:** 1Key Laboratory of Bio-Resource and Eco-Environment of Ministry of Education, College of Life Sciences, Sichuan University, Chengdu 610065, China; 2College of Food and Biological Engineering, Chengdu University, Chengdu 610106, China; 3Laboratory of Molecular Biology, College of Medicine, Chengdu University, Chengdu 610106, China; 4School of Pharmacy, Chengdu Medical College, Chengdu 610500, China; 5College of Pharmacy, Chengdu University of Traditional Chinese Medicine, Chengdu 611137, China; 6College of Medical Technology, Chengdu University of Traditional Chinese Medicine, Chengdu 611137, China; 7Microbiology and Metabolic Engineering Key Laboratory of Sichuan Province, College of Life Sciences, Sichuan University, Chengdu 610065, China

**Keywords:** myriocin, advanced glycation end products, obesity, AMPK, PGC1α

## Abstract

**Background**: Diet-derived advanced glycation end products (dAGEs) are closely associated with obesity and metabolic disorders. This study investigates the therapeutic potential of myriocin (Myr), a sphingolipid synthesis inhibitor, in counteracting dAGE-induced obesity and its underlying mechanisms. **Methods**: Male C57BL/6J wild-type mice were randomly assigned to receive either a low-AGE diet or a high-AGE diet with or without the administration of myriocin for a duration of 24 weeks. At the end of the experimental period, blood samples, whole livers, and adipose tissues were harvested for subsequent biochemical, histological, and molecular analyses. **Results**: Using a 24-week high-AGE diet mouse model, we demonstrate that Myr significantly reduces body weight gain (by 76%) and adipose tissue accumulation, while alleviating hepatic steatosis. Myr improves glucose homeostasis by lowering fasting blood glucose (a 44.5% reduction), enhancing oral glucose tolerance, and restoring hepatic glycolysis/gluconeogenesis balance via upregulating glucokinase and suppressing *G6pc*. Notably, Myr reduces serum LDL-C, TG, and TC levels by 52.3%, 51.8%, and 48.8%, respectively, and ameliorates liver dysfunction as evidenced by normalized ALT/AST activities. Metabolomics reveal Myr reshapes amino acid, carbohydrate, and lipid metabolism pathways. Mechanistically, Myr suppresses lipogenesis by downregulating *Srebp1*, *Fasn*, and *Acc*, while activating AMPK-PGC1α signaling to enhance mitochondrial biogenesis (a 2.1-fold increase in mtDNA) and thermogenesis via *Ucp1* upregulation in brown and white adipose tissues. **Conclusions**: Our findings unveil Myr as a novel dual regulator of lipid and glucose metabolism through AMPK-PGC1α-mediated mitochondrial activation, providing the first evidence of sphingolipid inhibition as a therapeutic strategy against dAGE-induced metabolic syndrome. This study establishes a multifaceted mechanism involving hepatic lipid regulation, adipose browning, and systemic metabolic reprogramming, advancing potential clinical applications for obesity-related disorders.

## 1. Introduction

Advanced glycation end products (AGEs), formed through non-enzymatic glycation during thermal food processing, have emerged as critical contributors to obesity and metabolic syndrome [1]. Epidemiological and experimental studies consistently link high dietary AGE (dAGE) intake to adipose tissue dysfunction, insulin resistance, and cardiovascular risk, mediated through pro-inflammatory pathways and oxidative stress [2,3,4,5]. Obesity, characterized by an excessive accumulation of adipose tissue, results from an imbalance in energy intake and expenditure, often exacerbated by environmental and dietary factors such as excessive fat and sugar intake. One of the key mechanisms underlying obesity is the dysfunction of white adipose tissue (WAT), which leads to excessive fat storage and chronic inflammation [6]. In contrast, brown adipose tissue (BAT), known for its thermogenic capacity, plays a protective role against obesity by dissipating energy as heat through the action of uncoupling protein 1 (UCP1) [7]. A key regulator of energy homeostasis, the AMP-activated protein kinase (AMPK)-peroxisome proliferator-activated receptor gamma coactivator-1α (PGC1α) axis, governs mitochondrial biogenesis and adipose browning [8,9,10]. Activation of this pathway enhances fatty acid oxidation and UCP1-mediated thermogenesis, offering a therapeutic target for metabolic disorders. However, pharmacological strategies to sustainably activate AMPK-PGC1α signaling in obesity models, particularly in dAGE-exposed models, remain underexplored.

Sphingolipids, particularly ceramides, are bioactive molecules intricately involved in metabolic regulation, with elevated levels implicated in insulin resistance, lipotoxicity, and mitochondrial dysfunction [11]. Ceramides disrupt insulin signaling by inhibiting AKT activation, promote adipose tissue inflammation through NF-κB pathways, and impair lipid oxidation in liver and muscle, thereby exacerbating obesity-related metabolic disturbances [12]. The pharmacological inhibition of sphingolipid synthesis, such as targeting serine palmitoyl transferase (SPT), the rate-limiting enzyme in de novo ceramide production, has emerged as a promising strategy to restore glucose homeostasis and lipid metabolism [13].

Myriocin (Myr), a potent inhibitor of serine palmitoyl transferase (SPT), suppresses de novo sphingolipid synthesis and has shown pleiotropic metabolic benefits, including an extension of the lifespan of yeast and the mitigation of muscle aging and ferroptosis in mammals [14,15,16,17]. Intriguingly, emerging evidence suggests that Myr modulates AMPK activity, but its role in dAGE-induced metabolic dysregulation and the involvement of AMPK-PGC1α signaling remain unknown. While sphingolipid reduction has been associated with improved insulin sensitivity and lipid metabolism [11,12,13], the systemic metabolic remodeling effects of Myr, particularly its capacity to restore mitochondrial function and adipose plasticity, have not been investigated in the context of dietary AGEs.

Here, we hypothesize that Myr ameliorates dAGE-induced obesity by activating the AMPK-PGC1α axis, thereby enhancing mitochondrial energetics and adipose browning. Using a 24-week high-AGE diet murine model, we integrate metabolomics, molecular profiling, and functional assays to identify Myr’s efficacy in reversing obesity and metabolic disorders and elucidate the AMPK-PGC1α-UCP1 cascade as the central mechanism driving these benefits. This study provides the first evidence of sphingolipid inhibition as a strategy to counteract dAGE-induced metabolic syndrome through mitochondrial activation and adipose tissue remodeling.

## 2. Materials and Methods

### 2.1. Reagents and Materials

Myriocin (APExBIO, Houston, TX, USA, B6064) and glucose (Chronchem, Chengdu, China) were used as reagents in the study. Various assay kits for the determination of serum biochemical markers were obtained from Nanjing Jiancheng Bioengineering Institute (Nanjing, China), including those for glycated serum proteins (GSP, A037-2-1), aspartate aminotransferase (AST, C010-2-1), alanine aminotransferase (ALT, C009-2-1), high-density lipoprotein cholesterol (HDL-C, A112-1-1), low-density lipoprotein cholesterol (LDL-C, A113-1-1), triglycerides (TG, A110-1-1), and total cholesterol (TC, A111-1).

### 2.2. Preparation of AGE Diet

The control (low-AGEs) diet was based on the standard AIN-93G formulation (HFK Bio-Science Co., Ltd., Beijing, China), which consists of 20% protein, 64% carbohydrate, 7% fat, and provides 3.9 kcal/g. High-AGE diets were prepared as previously described [18]. Specifically, the low-AGE and high-AGE diets were prepared using the same ingredients and formulations, with the high-AGE diet subjected to an additional heating step (165 °C for 1 h) with water compensation to facilitate the formation of advanced glycation end products.

### 2.3. Detection of AGEs Content

To quantify the levels of AGEs, 0.1 g of standard (low-AGE) and high-temperature-heated (high-AGE) feeds were each weighed and separately homogenized in 1 mL of phosphate-buffered saline (PBS). After centrifugation, 200 μL of the supernatant was collected for fluorescence analysis. The AGE content was determined by measuring the fluorescence emission at 440 nm following excitation at 370 nm, using a fluorescence spectrometer. The absorbance values for both the standard and heated feeds were calculated and compared.

### 2.4. Animal Experiment Design

Healthy male C57BL/6J wild-type mice, aged 8 weeks, were procured from the Center for Laboratory Animal Science, Sichuan University (Chengdu, China). Mice were housed under standard laboratory conditions with free access to tap water and a standard rodent chow diet, following the ethical guidelines of Sichuan University. Each group (n = 6) received either a low-AGE diet (control) or a high-AGE diet, with or without myriocin supplementation (2.2 mg/kg diet). This dosage was selected based on prior studies confirming its efficacy in inhibiting SPT activity without toxicity [19,20,21], and the intervention lasted 24 weeks. At the end of the experimental period, blood samples, whole livers, and adipose tissues were harvested for subsequent biochemical, histological, and molecular analyses. There were no criteria or exclusions. Blinding was not implemented in this study. Confounders were not controlled.

### 2.5. Determination of Body Weight, Adiposity Index, and Serum Biochemical Markers

Mice were weighed weekly for the duration of the 24-week study. At the end of the dietary treatment period, various adipose tissues were harvested and weighed. The adiposity index was calculated as the ratio of adipose tissue weight to total body weight.

### 2.6. Oral Glucose Tolerance Test (OGTT)

For the oral glucose tolerance test (OGTT), mice were fasted overnight for 12 h with free access to water and then orally gavaged with a 20% D-glucose solution (2.0 g/kg body weight for C57BL/6J mice). Blood glucose levels were measured at 0, 30, 60, 90, and 120 min post-gavage from the tail vein using an ACCU-CHEK Performa glucometer (Roche, Basel, Switzerland).

### 2.7. Determination of Serum Biochemical Markers

After 24 weeks of dietary intervention, blood samples were allowed to clot at room temperature for 1 h. The samples were then centrifuged at 4 °C at 3000 rpm for 10 min, and the resulting serum supernatant was collected for biochemical analysis. A portion of the serum was stored at −80 °C for subsequent metabolome analysis, with care taken to prevent multiple freeze/thaw cycles. For biochemical assays, 10 μL and 2.5 μL aliquots of serum were used to assess various markers, including glycated serum proteins (GSPs), triglycerides (TGs), total cholesterol (TC), high-density lipoprotein cholesterol (HDL-C), low-density lipoprotein cholesterol (LDL-C), alanine aminotransferase (ALT), aspartate aminotransferase (AST), and nitric oxide (NO), following the manufacturer’s instructions for the respective assay kits (Nanjing Jiancheng Bioengineering Institute, Nanjing, China).

### 2.8. Metabolite Extraction and LC-MS Analysis

For metabolomic analysis, six serum samples from each experimental group were selected for serum metabolome profiling. Metabolite extraction and analysis were performed in collaboration with Metabo Profile Technology Co., Ltd. (Shanghai, China). Liquid chromatography–mass spectrometry (LC-MS/MS) was conducted using an ultra-high-performance liquid chromatography (UHPLC) system (Thermo Fisher Scientific, Waltham, MA, USA) equipped with an ACQUITY UPLC^®^ HSS T3 column (1.8 µm, 2.1 mm × 150 mm) (Waters, Milford, MA, USA). The LC-MS data acquisition and processing protocols have been described in previous studies [22,23].

### 2.9. Pathological Histology Analyses

Liver tissues were fixed in 10% formalin for 48 h and then dehydrated through graded alcohol concentrations. The tissues were subsequently embedded in paraffin and sectioned into 5–6 µm slices. Hematoxylin and eosin (HE) staining was performed, and the specimens were examined under a light microscope at magnifications of 40× and 400× to assess histopathological changes.

For Oil Red O staining, non-fixed liver tissues were embedded in optimal cutting temperature (OCT) compound (Tissue-Tek, Torrance, CA, USA) and frozen in liquid nitrogen. Cryosections (10 µm thick) were prepared using a Thermo HM525 cryostat (Thermo Fisher Scientific, Waltham, MA, USA) and stored at −80 °C until use. Sections were stained with Oil Red O working solution (0.5% Oil Red O in isopropanol) for 30 min and counterstained with hematoxylin. 

### 2.10. Reverse Transcription Quantitative PCR (RT-qPCR) Analysis

Tissues were homogenized using a homogenizer (KZ-III-F; Servicebio, Wuhan, China), and total RNA was extracted using Beyozol reagent (Beyotime, Nantong, China, R0011) following the manufacturer’s instructions. The RNA concentration and purity were measured using a Nanodrop spectrophotometer. RNA was then reverse-transcribed into complementary DNA (cDNA) using an RNase-free cDNA synthesis kit (Beyotime, D7168M), following the manufacturer’s protocol. Primers for target genes were designed using the Nucleotide database (NCBI) and synthesized by Tsingke Biotechnology Co., Ltd. (Beijing, China). The sequences of the primers are provided in Table 1.

Quantitative PCR (qPCR) was performed using SYBR Green qPCR Master Mix (APExBIO, Houston, TX, USA, K1070) on a CFX96 Touch real-time PCR system (Bio-Rad, Hercules, CA, USA). The amplification conditions included an initial 2-min denaturation step at 95 °C, followed by 40 cycles of denaturation at 95 °C for 15 s, annealing at 60 °C for 30 s, and extension at 72 °C for 30 s. A final melting curve analysis was conducted with a 15 s hold at 95 °C, followed by a 4 °C hold. Each sample was run in triplicate. Actin beta (*Actb*) was used as the internal control to normalize the data, and relative gene expression levels were calculated using the ΔΔCq method.

### 2.11. Western Blotting Analysis

Indicated organs from the specified groups were homogenized in RIPA lysis buffer (CWBIO, Beijing China, CW2333S), containing 50 mM Tris-HCl pH 7.4, 150 mM NaCl, 1% Triton X-100, 1% sodium deoxycholate, 0.1% SDS, supplemented with sodium orthovanadate, sodium fluoride, EDTA, and leupeptin as proprietary additives. The buffer was further supplemented with protease inhibitor cocktail (APExBIO, K1007) and phosphatase inhibitor cocktail (APExBIO, K1012). Protein samples were separated using 8–12% SDS-PAGE gels and transferred onto polyvinylidene fluoride (PVDF) membranes (Millipore, Burlington, MA, USA, IPVH00010). The membranes were then blocked with 5% non-fat milk in 1× TBST (20 mM Tris, 150 mM NaCl, 0.1% Tween-20, pH 7.5) for 1 h at room temperature. Following blocking, the membranes were incubated overnight at 4 °C with primary antibodies: p-AMPK (Abways, Shanghai, China, #CY5608, 1:2000), AMPK (Abways, CY5326, 1:2000), UCP1 (Wanleibio, WL02625, 1:2000), PGC1α/β (Abways, CY6630, 1:2000), and β-Tubulin (Abmart, Shanghai, China, M30109S, 1:3000). The following day, membranes were washed three times with 1× TBST and incubated with the appropriate HRP-conjugated secondary antibodies (Beyotime, A0208, 1:3000; A0216, 1:3000) for 1 h at room temperature. Signals were detected using the BeyoECL Moon kit (Beyotime, P0018FM), and imaging and signal quantification were performed using the ChemiDocTMXRS^+^ imager (Bio-Rad).

Due to potential antibody cross-reactivity, limitations in sequential stripping (e.g., signal degradation), and overlapping migration positions of certain proteins (e.g., AMPK and Tubulin), distinct blots were used for separate antibody probes. To ensure consistency, replicate gels were loaded with identical sample quantities, and normalization was performed against Tubulin, which served as the shared loading control across experiments.

### 2.12. Statistical Analysis

Data are expressed as mean ± standard deviation (SD). GraphPad Prism 9.0 was used for data visualization and statistical analysis. For comparisons between two groups, an unpaired Student’s *t*-test was used, while two-way ANOVA was applied for multivariate analysis. Statistical significance was determined as follows: ns, *p* > 0.05; * *p* < 0.05; ** *p* < 0.01; *** *p* < 0.001; **** *p* < 0.0001.

## 3. Results

### 3.1. Myr Prevents Obesity Induced by dAGEs

To establish a diet-induced obesity model with elevated advanced glycation end products (AGEs), standard AIN-93G chow was subjected to high-temperature baking. Fluorescence spectrophotometry confirmed that the AGE content in the heated chow was 3.8-fold higher than in the standard chow (Figure 1A). Mice were divided into four groups: control diet (L-AGEs), control diet + Myr (L-AGEs + Myr), high-AGE diet (H-AGEs), and high-AGE diet + Myr (H-AGEs + Myr). Over 24 weeks, their body weight and food intake were monitored weekly (Figure 1B). While the initial body weights were comparable between groups, a significant divergence emerged after 10 weeks (Figure 1C). By week 24, the H-AGEs group exhibited a 76% greater weight gain compared to the control group (Figure 1D). Strikingly, Myr supplementation in the H-AGEs + Myr group attenuated this weight gain, demonstrating its anti-obesity efficacy. Notably, although the H-AGEs + Myr group showed a marginal reduction in average food intake relative to the H-AGEs group, this difference lacked statistical significance (Figure 1E), indicating that Myr counteracts dAGE-induced obesity independently of appetite suppression.

In addition to body weight, we assessed adipose tissue weight, a key indicator of obesity [24]. The H-AGE diet significantly increased the weight of inguinal white adipose tissue (iWAT) and epididymal white adipose tissue (eWAT), while only minor changes were observed in brown adipose tissue (BAT) (Figure 1F). Myr treatment reversed the increase in adipose tissue weight induced by dAGEs (Figure 1F). Collectively, these findings demonstrate that Myr mitigates dAGE-induced weight gain by preventing excessive fat accumulation.

### 3.2. Myr Improves Glucose Homeostasis in Mouse Group Fed High AGE Diet

Given the established link between dAGEs and diabetes risk, we investigated Myr’s impact on glucose metabolism. Fasting blood glucose (FBG) levels in the H-AGEs group rose to 10.3 ± 0.31 mmol/L, representing a 44.5% increase compared to the control group (Figure 2A). Notably, impaired glucose tolerance, defined by 2 h postprandial glucose levels exceeding 7.8 mmol/L [25], was evident in the H-AGEs group, with sustained hyperglycemia (10.25 mmol/L at 120 min) during the oral glucose tolerance test (OGTT) (Figure 2B). In contrast, Myr supplementation significantly reduced both FBG levels (Figure 2A) and blood glucose concentrations at 60 and 90 min during the OGTT (Figure 2B). Furthermore, Myr treatment significantly reduced the area under the curve (AUC) for the OGTT (Figure 2C).

Additionally, we measured glycated serum protein (GSP) levels, a marker of chronic hyperglycemia. The H-AGEs group exhibited significantly higher GSP levels compared to the L-AGEs group, while Myr treatment reduced GSP levels by 9.8%, although treatment with Myr for the L-AGEs group unexpectedly increased GSP levels (Figure 2D).

Glucose homeostasis is primarily regulated by hepatic glycolysis and gluconeogenesis. The glycolytic pathway involves the breakdown of glycogen and glucose into pyruvate, with glucokinase (Gck) serving as a key regulatory enzyme. Conversely, gluconeogenesis synthesizes glucose from non-carbohydrate precursors, with glucose-6-phosphatase (G6pc) catalyzing the terminal step of glucose-6-phosphate dephosphorylation. Here, we investigated the hepatic expression of these rate-limiting enzymes. The H-AGE diet suppressed hepatic *Gck* mRNA levels while upregulating *G6pc* expression (Figure 2E,F). The Myr intervention counteracted these effects, increasing *Gck* expression by 1.8 fold and reducing *G6pc* by 62% (Figure 2E,F). These molecular changes correlate with enhanced hepatic glucose utilization and diminished gluconeogenic output, synergistically improving systemic glycemic control.

### 3.3. Myr Ameliorates Ectopic Lipid Deposition in the Liver of Mice Fed High AGE Diet

We next assessed the impact of Myr on hepatic lipid metabolism by measuring serum lipid levels, including low-density lipoprotein cholesterol (LDL-C), high-density lipoprotein cholesterol (HDL-C), triglycerides (TG), and total cholesterol (TC). Myr supplementation in the H-AGEs + Myr group significantly attenuated serum LDL-C, TG, and TC by 52.3%, 51.8%, and 48.8%, respectively, while hepatic TG and TC decreased by 65.8% and 58.2% (Figure 3A,D). Notably, serum high-density lipoprotein cholesterol (HDL-C) increased by 68.8% (Figure 3A), suggesting the selective regulation of Myr in lipid metabolism pathways.

Hepatic dysfunction, a hallmark of lipid overload, was further corroborated by elevated serum and hepatic alanine aminotransferase (ALT) and aspartate aminotransferase (AST) activities in the H-AGEs group (Figure 3B,C). Myr intervention restored ALT and AST levels to values close to those of the control (Figure 3B,C), suggesting that Myr helps restore liver function.

To examine the effects of Myr on liver histopathology, we performed hematoxylin and eosin (HE) staining. The mice fed H-AGEs displayed an indistinct hepatic sinusoidal structure, a loose hepatocyte structure, and signs of edema and vacuolar degeneration, whereas the Myr-treated mice exhibited a radial arrangement of hepatic sinusoids along the central vein, a well-defined hepatocyte structure, and minimal signs of vacuoles (Figure 3E, upper panels).

We further evaluated hepatic lipid accumulation using Oil Red O staining. In the mice fed H-AGEs, there was extensive accumulation of lipid droplets in liver tissue, as evidenced by bright red staining, which was significantly reduced by the Myr treatment (Figure 3E, lower panels). These findings, along with the reduction in hepatic TG and TC levels (Figure 3D), indicate that Myr mitigates lipid accumulation in the liver induced by dAGEs.

### 3.4. Myr Treatment Alters Serum Metabolite Profile in Mice Fed dAGEs

To explore the broader metabolic effects of Myr on mice fed H-AGEs, we performed an untargeted metabolomics analysis of serum samples from these mice. Principal component analysis (PCA) of the serum metabolites revealed the clear clustering of samples based on treatment groups (Figure 4A), indicating substantial differences in the metabolite profiles between the Myr-treated and untreated groups. The orthogonal partial least squares discriminant analysis (OPLS-DA) model further confirmed these differences, showing a distinct separation of the groups in both positive and negative ion modes (Figure 4B).

Using criteria such as a variable importance in projection (VIP) value greater than 1.0, fold changes greater than 1.0 or less than 1.0, and a *p*-value less than 0.05, we identified 62 significantly altered metabolites (Figure 4C). These metabolites were predominantly involved in amino acid metabolism, carbohydrate metabolism, and lipid metabolism, as confirmed by Kyoto Encyclopedia of Genes and Genomes (KEGG) pathway annotation (Figure 4D). Specific pathways included in the involved metabolisms were histidine metabolism, central carbon metabolism in cancer, bile secretion, protein digestion and absorption, tryptophan metabolism, the cAMP signaling pathway, and more (Figure 4E).

Overall, these results indicate that Myr treatment induces significant shifts in the serum metabolite profile, particularly in pathways related to energy metabolism, amino acid regulation, and lipid processing. This metabolic transition likely contributes to Myr’s beneficial effects of improving obesity and glucose homeostasis in mice fed H-AGEs.

### 3.5. Myr Regulates Factors Related to Lipid Metabolism in Liver

Given the pronounced effects of Myr on hepatic lipid metabolism, we next analyzed the expression of genes central to lipid regulation. Myr significantly reduced the mRNA levels of sterol regulatory element-binding protein 1 (*Srebp1*), a master regulator of lipid homeostasis and glucose metabolism [26,27], in the livers of high-AGE diet mice (Figure 5). Furthermore, the H-AGE diet markedly upregulated lipogenic genes, including acetyl-CoA carboxylase (*Acc1*), fatty acid synthase (*Fasn*), and peroxisome proliferator-activated receptor gamma (*Pparg*), which were effectively suppressed by Myr intervention. Conversely, the H-AGE diet suppressed the expression of lipolytic genes—hormone-sensitive lipase (*Hsl*) and adipose triglyceride lipase (*Atgl*)—whereas Myr treatment restored their expression (Figure 5). These findings collectively suggest that Myr counteracts dAGE-induced hepatic lipogenesis by suppressing SREBP1-mediated transcriptional activation, thereby mitigating lipid overaccumulation.

### 3.6. Myr Promotes Adipose Thermogenesis in Mice Fed H-AGEs

In addition to inducing liver and systemic changes in metabolisms, decreased body and adipose weight suggest that Myr may have profound effects on adipose tissue. To investigate the impact of Myr on adipose tissue remodeling, we first examined morphological changes in brown (BAT) and inguinal white adipose tissue (iWAT). Histological analysis revealed that mice fed H-AGEs exhibited hypertrophic adipocytes in both BAT and iWAT, whereas Myr intervention restored adipocyte size to a near-normal morphology (Figure 6A). This phenotypic reversal suggested enhanced thermogenic activity, prompting us to evaluate the expression of uncoupling protein 1 (UCP1), a hallmark of thermogenesis. Myr treatment significantly upregulated *Ucp1* mRNA levels in BAT and iWAT compared to the H-AGEs group (Figure 6B), a finding corroborated by elevated UCP1 protein expression via Western blotting (Figure 6C).

To elucidate the underlying regulatory mechanisms, we analyzed thermogenic markers in both BAT and iWAT. Myr treatment in mice fed with either L-AGEs or H-AGEs significantly increased Type 2 iodothyronine deiodinase (*Dio2*) mRNA levels in BAT (Figure 6D), a key regulator of intracellular triiodothyronine conversion that is essential for *Ucp1* activation. Simultaneously, Myr administration in mice fed H-AGEs elevated cell death-inducing DNA fragmentation factor alpha subunit-like effector A (*Cidea*) expression in iWAT (Figure 6E), a marker of adipose tissue browning that promotes lipid droplet remodeling and mitochondrial uncoupling. The upregulation of *Dio2* in BAT and *Cidea* in iWAT highlights Myr’s ability to enhance thermogenesis through distinct, yet complementary, pathways.

### 3.7. Myr Activates the AMPK-PGC1α Signaling Pathway

Mitochondria play a critical role in regulating energy metabolism by controlling energy expenditure and thermogenesis, particularly in brown adipose tissue (BAT). Mitochondrial biogenesis is primarily regulated by factors such as PGC1α, NRF1, and TFAM [28]. As a transcriptional coactivator, PGC1α governs mitochondrial DNA (mtDNA) replication and the expression of genes involved in mitochondrial function, including NRF1 and TFAM. We investigated the effects of Myr on mitochondrial biogenesis by analyzing the mRNA levels of *Pgc1a*, *Pgc1b*, *Nrf1*, *Nrf2*, and *Tfam* in various tissues. Our results showed that Myr treatment significantly increased the mRNA levels of these genes in the liver (Figure 7A). In BAT, Myr enhanced *Pgc1a*, *Pgc1b*, and *Nrf1*, although it had no effect on *Nrf2* and *Tfam* expression. In iWAT, Myr upregulated *Pgc1a*, *Pgc1b*, and *Nrf1* expression, while *Tfam* levels remained unaffected (Figure 7A). These data suggest that Myr treatment induced the systemic activation of mitochondrial function.

We also measured the mtDNA copy number, which reflects mitochondrial biogenesis. Myr treatment did not alter mtDNA levels in iWAT but significantly increased mtDNA copy numbers in both BAT and the liver (Figure 7B). This suggests that Myr promotes mitochondrial biosynthesis, particularly in BAT and the liver.

Further, Myr treatment led to a significant increase in PGC1α protein expression in BAT, iWAT, and the liver, compared to the H-AGEs group (Figure 7C). Since PGC1α expression is regulated by the energy sensor AMPK [8], we next assessed AMPK activation by measuring the phosphorylation levels of AMPK and PGC1α. Myr treatment significantly increased the phosphorylation of AMPK and PGC1α in the liver, BAT, and iWAT (Figure 7C). These findings suggest that Myr activates the AMPK-PGC1α signaling pathway, which may play a key role in regulating mitochondrial homeostasis and energy expenditure.

## 4. Discussion

Obesity, characterized by excess body fat, is intricately linked to disturbances in lipid metabolism, particularly in WAT and BAT. In this study, we investigated the effects of Myr, a sphingolipid synthesis inhibitor, on diet-induced obesity in mice exposed to advanced glycation end products. Our results demonstrate that Myr mitigates obesity and hyperglycemia by improving lipid metabolism, enhancing mitochondrial biogenesis, and promoting thermogenesis via the AMPK-PGC1α signaling pathway.

Our findings align with emerging evidence on sphingolipid modulation in metabolic regulation and expand our mechanistic understanding of it in the context of dietary AGE-induced obesity. It was demonstrated that Myriocin restores mitochondrial proteostasis in aged skeletal muscle by reducing ceramide synthesis [17], mirroring our observation of enhanced mitochondrial biogenesis in liver and adipose tissues through AMPK-PGC1α activation. While other works have focused on muscle aging, our data reveal a broader systemic effect of Myriocin. Notably, Chaurasia et al. (2016) [29] and Handzlik et al. (2023) [30] established that ceramide depletion promotes adipose browning and insulin sensitization, consistent with our findings of Myr-induced *Cidea* upregulation in iWAT and UCP1 activation in BAT. However, our study uncovers two critical advancements: First, we identify dietary AGEs as a novel metabolic stressor that exacerbates ceramide-driven mitochondrial dysfunction, a dimension absents in previous models. Second, while works have implicated PPARγ and PRDM16 in adipose remodeling, we provide the first evidence that Myr’s effects are mechanistically rooted in AMPK-PGC1α axis activation. PPARγ and PRDM16 form a transcriptional complex that recruits PGC1α to activate thermogenic and mitochondrial genes, driving oxidative metabolism programs and brown/beige adipocyte differentiation [31,32]. However, how Myr-induced AMPK activation modulates these factors and their interactions warrants further investigation.

We demonstrate that Myr suppresses lipogenic gene expression (e.g., Srebp1, Acc, and Fasn) and enhances lipid catabolism through the transcriptional regulation of Hsl and Atgl. However, while our data focus on mRNA-level changes, it is important to acknowledge that the activity of key enzymes such as ACC, HSL, and ATGL is tightly regulated by post-translational modifications (PTMs), which were not directly assessed in this study. For example, ACC activity is critically modulated by AMPK-mediated phosphorylation at Ser79, which inhibits its function in fatty acid synthesis [33]. Similarly, HSL and ATGL undergo phosphorylation and other PTMs that govern their lipolytic activity [34,35]. Thus, while our results indicate that Myr reduces hepatic Acc1 mRNA levels and upregulates Hsl and Atgl transcription, future studies are warranted to investigate whether Myr further modulates ACC, HSL, and ATGL activity through PTMs. Such work could elucidate whether sphingolipid inhibition synergizes with AMPK signaling to amplify metabolic benefits via dual transcriptional and post-translational regulation of these enzymes.

The activation of AMPK has a multifaceted impact, encompassing the regulation of both synthesis and catabolism, as well as the maintenance of mitochondrial homeostasis. PGC1α stands out as a pivotal protein in governing mitochondrial synthesis, with its expression also subject to regulation by AMPK. Research has demonstrated that AMPK can elevate the protein levels of PGC1α [36]. Activated PGC1α, in turn, can bind to the promoters of thermogenic genes, including UCP1, and collaboratively activate their transcription. Here, we found that inhibiting the de novo synthesis of sphingolipids by Myr drastically enhances the expression of UCP1. UCP1 uncouples mitochondrial respiration to generate heat, increasing energy expenditure [7]. This process reduces lipid accumulation, counteracts obesity, and induces WAT-to-beige remodeling (browning) via β-adrenergic signaling, cold exposure, or exercise [7,37]. The activation of classical brown fat or the browning of white fat has been shown to have a positive impact on obesity and glucose homeostasis in the body [38]. Additionally, it has been observed that increasing the browning of adipose tissue (BAT) can lead to a reduction in plasma TG and TC levels, contributing to the improvement of dyslipidemia [29,39]. Consistent with these findings, the current study indicates that Myr intervention activates the AMPK-PGC1α-UCP1 pathway in the liver and adipose tissue of mice. This results in an acceleration of energy expenditure, reduced serum and hepatic lipid levels, including LDL-C, TG, and TC, and restored liver function, as evidenced by decreased ALT and AST activities.

Future studies should incorporate pharmacological or genetic interventions targeting the AMPK pathway to confirm that the mechanism by which Myr influences obesity is indeed mediated through AMPK activation. Furthermore, additional research is required to elucidate the precise mechanism by which AMPK is activated following the inhibition of sphingolipid synthesis by Myr.

## 5. Conclusions

In summary, our study provides new insights into the therapeutic potential of Myr for treating obesity and metabolic disorders linked to excessive dietary AGEs. By activating the AMPK-PGC1α signaling pathway, Myr enhances mitochondrial function and thermogenesis, and improves lipid and glucose metabolism. The AMPK-PGC1α axis regulated by Myr emerges as a central node for counteracting dAGE-induced dysmetabolism, offering a multifaceted strategy against obesity and its complications.

## Figures and Tables

**Figure 1 nutrients-17-01549-f001:**
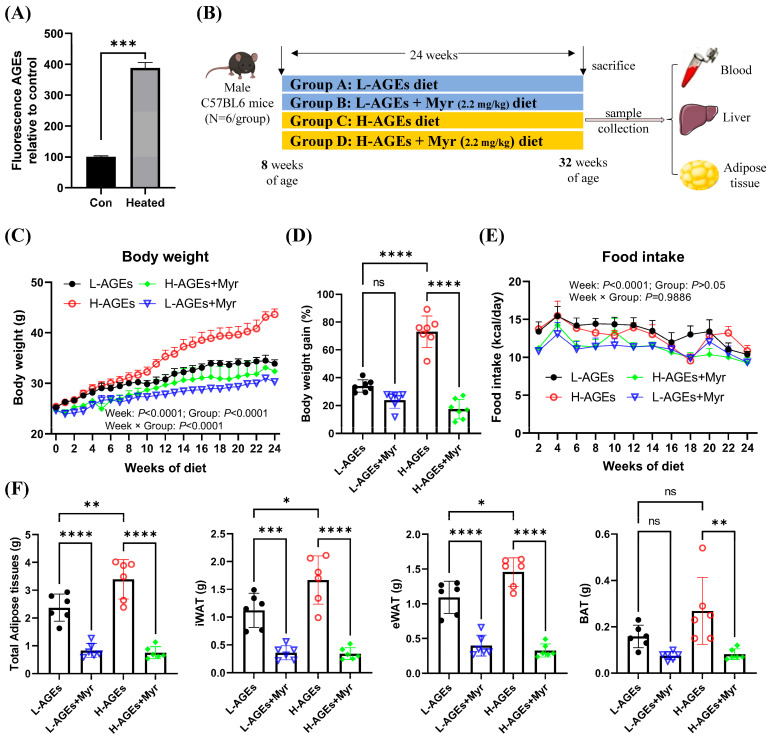
Myriocin prevented obesity in dAGE-fed mice. (**A**) AGE level in control and heated diet. (**B**) Experimental design chart. (**C**) Body weight, (**D**) body weight gain (%), and (**E**) food intake. (**F**) Adipose tissue weight; iWAT, inguinal white adipose tissue; eWAT, epididymal white adipose tissue; BAT, brown adipose tissue. Data are shown as mean ± SD (n = 6) (ns, *p* > 0.05, * *p* < 0.05, ** *p* < 0.01, *** *p* < 0.001, **** *p* < 0.0001), and were analyzed by *t*-test. (**C**,**E**) were analyzed by two-way ANOVA.

**Figure 2 nutrients-17-01549-f002:**
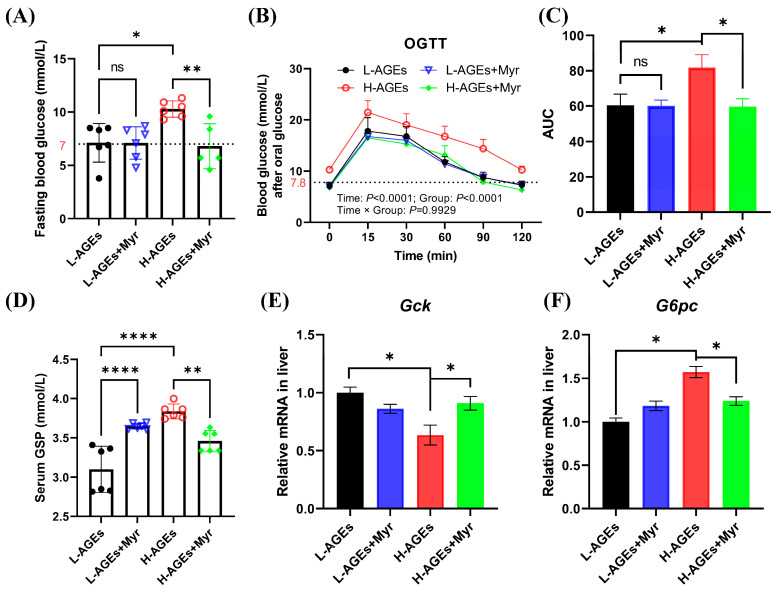
Myriocin improved glucose homeostasis in mice fed dAGEs. (**A**) Level of fasting blood glucose. (**B**) Oral glucose tolerance test (OGTT). (**C**) The area under curve (AUC) from the OGTT. (**D**) Level of serum glycosylated serum protein (GSP). (**E**,**F**) The mRNA expression of glucose metabolism in liver. *Gck*, glucokinase; *G6pc*, Glucose-6-phosphatase catalytic subunit. Data are shown as mean ± SD (n = 6) (ns, *p* > 0.05, * *p* < 0.05, ** *p* < 0.01, **** *p* < 0.0001) and were analyzed by *t*-test. (**B**) were analyzed by two-way ANOVA. The dotted lines in (**A**,**B**) represent baseline blood glucose levels.

**Figure 3 nutrients-17-01549-f003:**
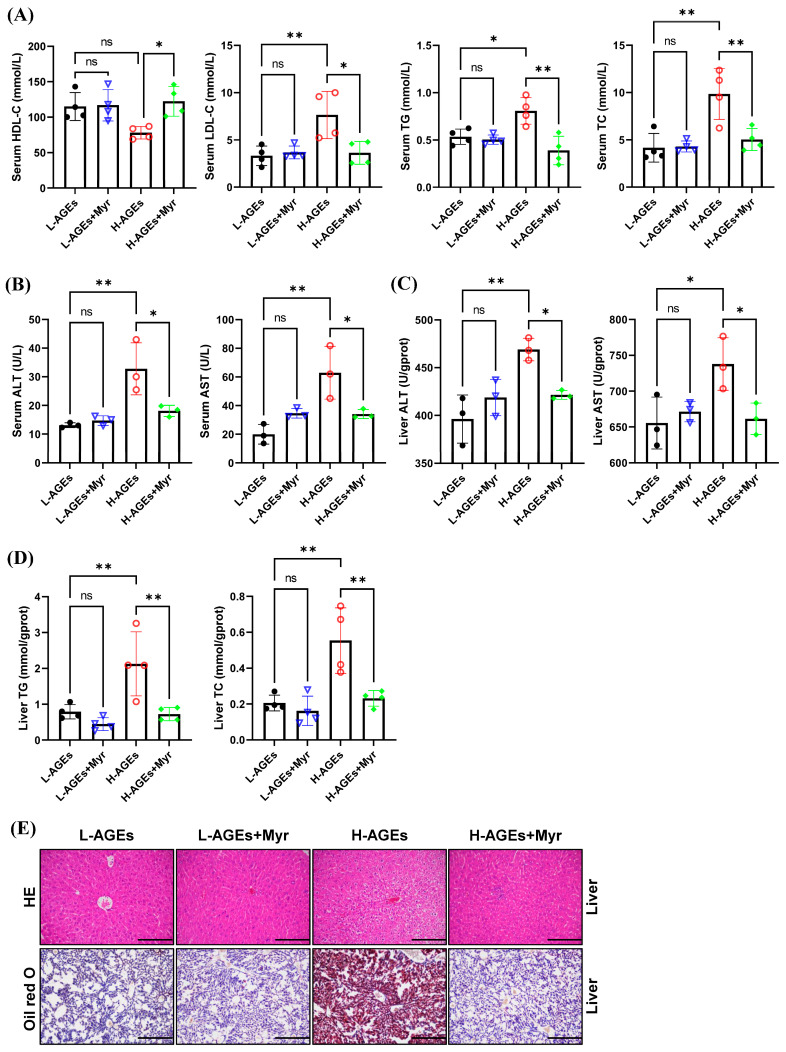
Myriocin ameliorated ectopic lipid deposition in mice fed dAGEs. (**A**) Serum lipid levels. ALT and AST levels in serum (**B**) and in liver (**C**). (**D**) Content of TG and TC in liver. (**E**) Representative images of HE and Oil red O staining of liver. 200×. ALT, Alanine transaminase; AST, aspartate aminotransferase; HE, hematoxylin and eosin; HDL-C, high-density lipoprotein cholesterol; LDL-C, low-density lipoprotein cholesterol; TG, triglyceride; TC, total cholesterol. Data are shown as mean ± SD (n = 3 or 4) (ns, *p* > 0.05, * *p* < 0.05, ** *p* < 0.01) and were analyzed by *t*-test.

**Figure 4 nutrients-17-01549-f004:**
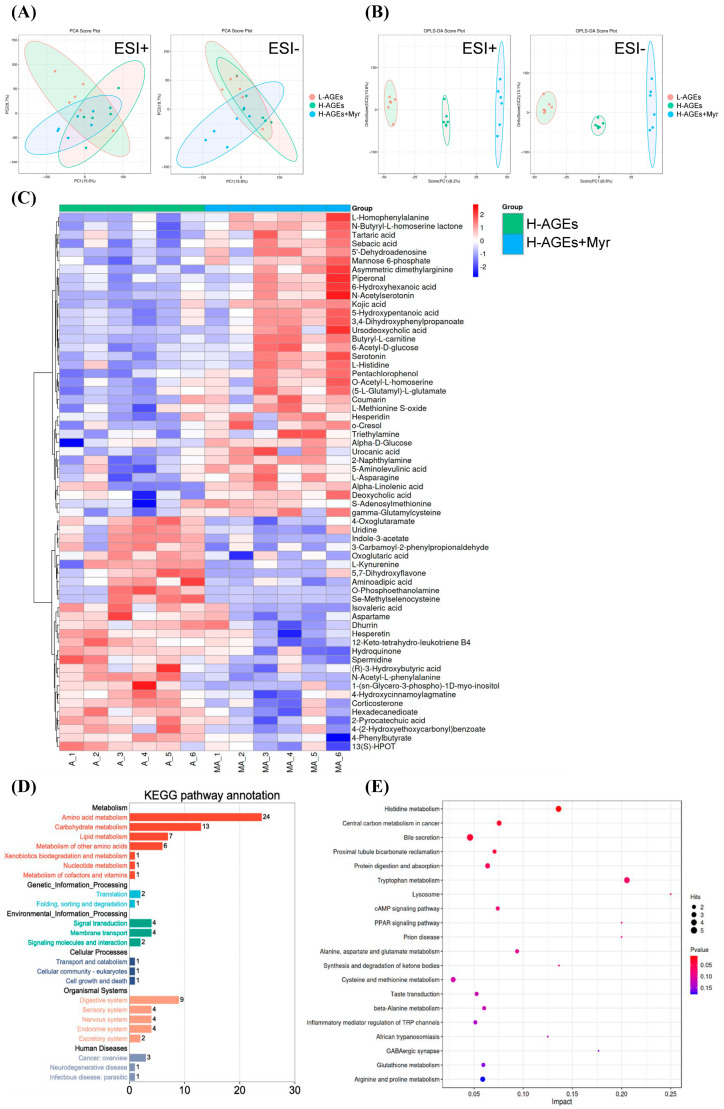
Myriocin regulated the serum metabolite profile in dAGEs-fed mice. PCA score plots (**A**) and OPLS-DA score plots (**B**) of serum samples from different groups; Heatmap (**C**) of serum differential metabolites in H-AGEs + Myr group compared to H-AGEs group. (**D**,**E**) represent KEGG annotation analysis of the changed metabolites and KEGG pathway analysis.

**Figure 5 nutrients-17-01549-f005:**
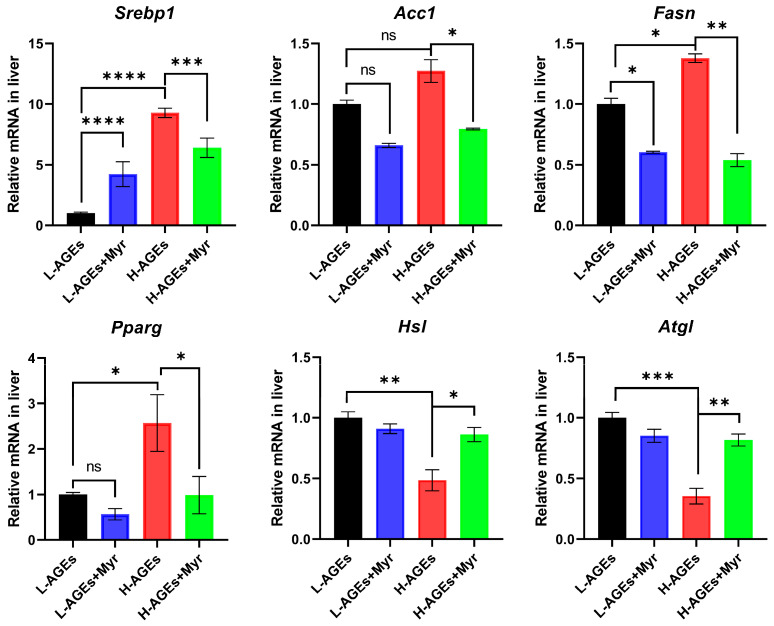
The effect of myriocin on mRNA expression of *Srebp1*, *Acc*, *Fasn*, *Pparg*, *Hsl*, and *Atgl* related to lipid metabolism in liver. The mRNA levels of lipogenic or lipolytic genes in liver were examined by RT-qPCR and normalized to *Actb*. Data are shown as mean ± SD (n = 3) (ns, *p* > 0.05, * *p* < 0.05, ** *p* < 0.01, *** *p* < 0.001, **** *p* < 0.0001) and were analyzed by *t*-test.

**Figure 6 nutrients-17-01549-f006:**
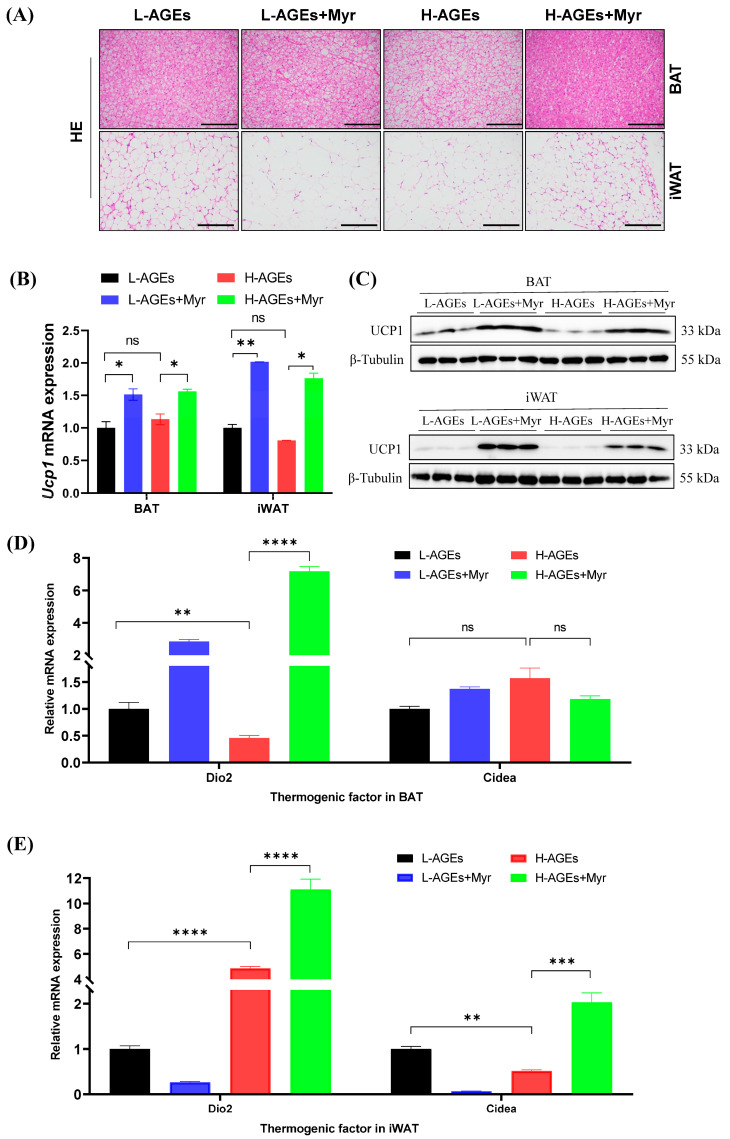
Myriocin promoted BAT thermogenesis and WAT browning in mice fed dAGEs. (**A**) Representative images of HE of BAT and iWAT. (**B**) The *Ucp1* mRNA levels in BAT and iWAT were analyzed by RT-qPCR. (**C**) Protein expressions of UCP1 in BAT and iWAT were analyzed by Western blotting, and β-Tubulin was used as the loading control. Separate blots were used for distinct antibodies due to technical constraints (see Methods). The mRNA levels of thermogenic genes in BAT (**D**) and iWAT (**E**) were examined by RT-qPCR and normalized to *Actb*. Data are shown as mean ± SD (n = 3) (ns, *p* > 0.05, * *p* < 0.05, ** *p* < 0.01, *** *p* < 0.001, **** *p* < 0.0001) and were analyzed by *t*-test.

**Figure 7 nutrients-17-01549-f007:**
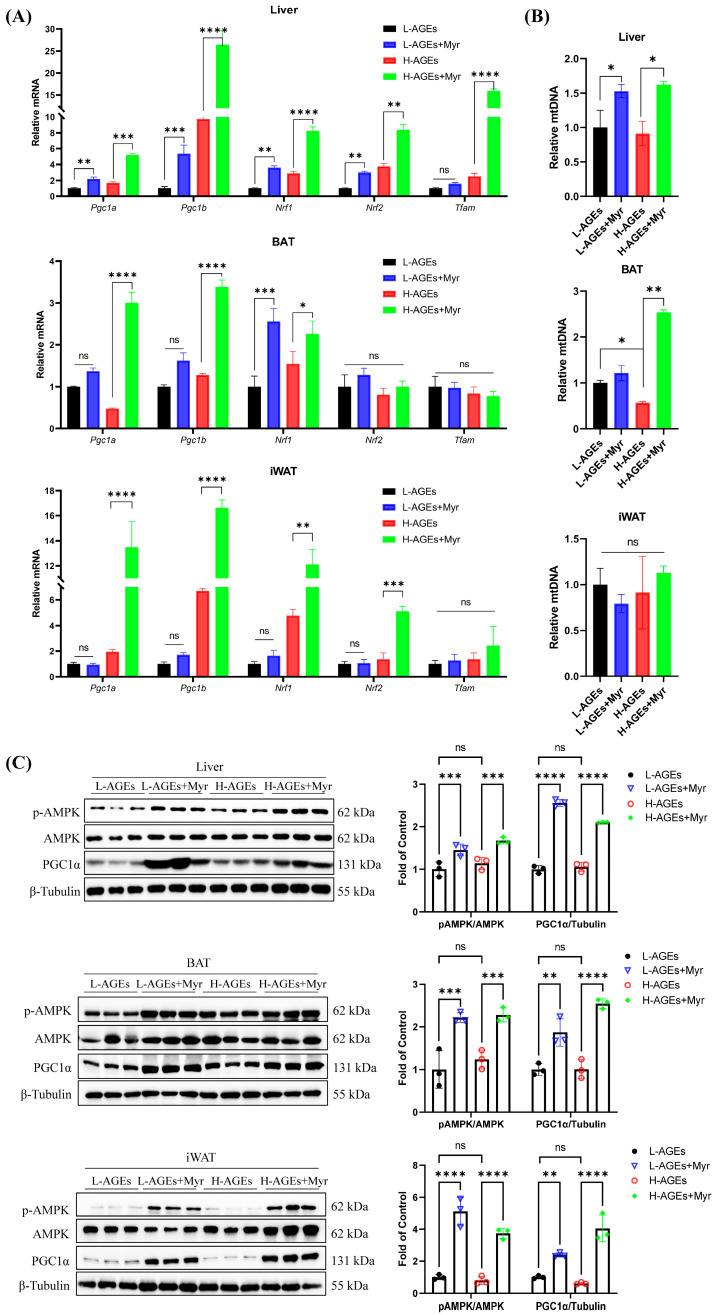
Myriocin-activated AMPK/PGC1α pathway in dAGE-fed mice. (**A**) The mRNA levels of genes involved in mitochondrial biogenesis in liver and adipose tissues were analyzed by RT-qPCR. (**B**) The mtDNA content calculated as the ratio of *Cox2* to *Actin* DNA levels measured by RT-qPCR in liver and adipose tissues. (**C**) The levels of p-AMPKα, AMPKα, and PGC1α in liver and adipose tissues were analyzed by Western blotting. Separate blots were used for distinct antibodies due to technical constraints (see Methods). β-Tubulin was used as the loading control. BAT, brown adipose tissue; iWAT, inguinal WAT. Data are shown as mean ± SD (n = 3) (ns, *p* > 0.05, * *p* < 0.05, ** *p* < 0.01, *** *p* < 0.001, **** *p* < 0.0001) and were analyzed by *t*-test.

**Table 1 nutrients-17-01549-t001:** Sequences of primers used in RT-qPCR.

Genes	Primer Pairs (5′-3′)	Protein Names
Actb	GATGTATGAAGGCTTTGGTC TGTGCACTTTTATTGGTCTC	Actin
Gck	CAACTGGACCAAGGGCTTCAA TGTGGCCACCGTGTCATTC	Glucokinase
G6pc	CAAGGGAGAACTCAGCAAGT GGGCTTCAGAGAGTCAAAGA	Glucose-6-phosphatase catalytic subunit 1
Srebp1c	GGAGCCATGGATTGCACATT GGCCCGGGAAGTCACTGT	Sterol regulatory element-binding protein 1
Acc1	CCCGCTCCTTCAACTTGCT ATTGGGCACCCCAGAGCTA	Acetyl-CoA carboxylase 1
Fasn	TGCTCCAGGGATAACAGC CCAAATCCAACATGGGACA	Fatty acid synthase
Pparg	CTTGCTGTGGGGATGTCT GGGTTCAGCTGGTCGATA	Peroxisome proliferator-activated receptor gamma
Hsl	GAGTAGTAACAAAGGTCAA CACAGTGACAGCCACATTCT	Hormone-sensitive lipase
Atgl	CTGGTCATCATCCTGCCTTT TTTTGGCAGAGGGAAAAAGA	Patatin-like phospholipase domain-containing protein
Ucp1	ACTGCCACACCTCCAGTCATT CTTTGCCTCACTCAGGATTGG	Mitochondrial brown fat uncoupling protein 1
Dio2	CAGTGTGGTGCACGTCTCCAATC TGAACCAAAGTTGACCACCAG	Type II iodothyronine deiodinase
Cidea	GCCGTGTTAAGGAATCTGCTG TGCTCTTCTGTATCGCCCAGT	Lipid transferase CIDEA
Pgc1a	CCCTGCCATTGTTAAGACC TGCTGCTGTTCCTGTTTTC	Peroxisome proliferator-activated receptor gamma coactivator 1-alpha
Pgc1b	TCCTGTAAAAGCCCGGAGTAT GCTCTGGTAGGGGCAGTGA	Peroxisome proliferator-activated receptor gamma coactivator 1-beta
Nrf1	GCTTCAGAACTGCCAACCAC TGTTCCACCTCTCCATCAGC	Nuclear respiratory factor 1
Nrf2	TAGATG ACCATGAGTCGCTTGC GCCAAACTTGCTCCATGTCC	Nuclear factor erythroid 2-related factor 2
Tfam	CAGGAGGCAAAGGATGATTC CCAAGACTTCATTTCATTGTCG	Transcription factor A, mitochondrial
Cox2	ACCAATAGCCCTGGCCGTAC GGTGGCGCTTCCAATTAGGT	Cytochrome c oxidase subunit 2

## Data Availability

The original contributions presented in this study are included in the article. Further inquiries can be directed to the corresponding authors.

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
