# Peer review of "Myriocin Restores Metabolic Homeostasis in dAGE-Exposed Mice via AMPK-PGC1α-Mediated Mitochondrial Activation and Systemic Lipid/Glucose Regulation"

_nutrients, 2025, doi:10.3390/nu17091549_

Round 1

Reviewer 1 Report

Comments and Suggestions for Authors

In the manuscript nutrients-3589972, the authors have demonstrated that Myriosin supplementation improved lipid and glucose metabolism in mice exposed to diet-derived advanced glycation end products (dAGE). This study has strength in measuring multiple indicators in serum, liver, and multiple depots of adipose tissue; however, there are some concerns and issues required to be addressed.

Major concerns

  1. Could the authors provide how the dose of Myriosin has been decided? Are there any scientific rationale?
  2. For this experimental design (2 factors: dietary AGE contents and Myr supplementation), two-way ANOVA is more suitable for the statistical analysis rather than one-way ANOVA.
  3. I wonder if there are remaining tissues for additional experiments. In Figure 5, to demonstrate the activation of ACC, HSL, and ATGL, the phosphor form of the proteins should be measured because their activity is regulated by phosphorylation.

Minor issues

  1. In the introduction part, explanation about the association between sphingolipid metabolism and lipid/glucose regulation would be helpful.
  2. In Table 1, adding a column showing gene function would be helpful.

Author Response

For research article

Response to Reviewer 1 Comments

1. Summary

2. Questions for General Evaluation

Reviewer’s Evaluation

Response and Revisions

Does the introduction provide sufficient background and include all relevant references?

Can be improved

See the point-by-point response blow

Is the research design appropriate?

Yes

Are the methods adequately described?

Yes

Are the results clearly presented?

Can be improved

See the point-by-point response blow

Are the conclusions supported by the results?

Yes

3. Point-by-point response to Comments and Suggestions for Authors

Comments and Suggestions for Authors

In the manuscript nutrients-3589972, the authors have demonstrated that Myriosin supplementation improved lipid and glucose metabolism in mice exposed to diet-derived advanced glycation end products (dAGE). This study has strength in measuring multiple indicators in serum, liver, and multiple depots of adipose tissue; however, there are some concerns and issues required to be addressed.

Major concerns:

Comments 1: Could the authors provide how the dose of Myriosin has been decided? Are there any scientific rationale?

Response 1: Thank you for highlighting the need for clarity regarding the Myriocin dosage. The selected Myriocin dosage (2.2 mg/kg diet, equivalent to ~0.4 mg/kg body weight/day) aligns with established protocols in murine studies, where it effectively inhibits serine palmitoyltransferase (SPT) activity without adverse effects (J Lipid Res . 2008 Feb;49(2):324-31; Proc Natl Acad Sci U S A. 2020 May 12;117(19):10565-10574; Cell Metab. 2025 Jan 7;37(1):274-290.e9). We acknowledge the inadvertent error in the original Figure 1B, where the supplementation of myriocin was incorrectly labeled as "(2.2 mg/kg/day)". This has been revised to "(2.2 mg/kg)". The rationale for dosage selection, including respective references, has been added to Section 2.4 (Animal experiment design).

Comments 2: For this experimental design (2 factors: dietary AGE contents and Myr supplementation), two-way ANOVA is more suitable for the statistical analysis rather than one-way ANOVA.

Response 2: We sincerely thank the reviewer for raising this critical point. You are absolutely correct that two-way ANOVA is the appropriate statistical method for analyzing the interaction between the two independent variables (dietary AGE content and Myr treatment). In our revised manuscript, we have carefully re-examined and clarified the statistical approaches used. For multi-group comparisons involving both factors (e.g., Figures 1C, 1E, 2B), two-way ANOVA​ was applied to assess main effects (diet, treatment) and their interaction. For single-variable comparisons (e.g., endpoint measurements in Figure 1D, 1F, 2A, etc.), unpaired Student’s t-test​ was applied for direct comparisons between two groups (e.g., H-AGEs vs. H-AGEs+Myr). The ​figure legends​ (e.g., Figures 1, 2, 3, 4, 6, 7) and ​Section 1.12 (Statistical Analysis)​​ have been revised to explicitly state the statistic method.

Comments 3: I wonder if there are remaining tissues for additional experiments. In Figure 5, to demonstrate the activation of ACC, HSL, and ATGL, the phosphor form of the proteins should be measured because their activity is regulated by phosphorylation.

Response 3: Thank you for this insightful suggestion. We agree that assessing phosphorylation status (e.g., phospho-ACC, phospho-HSL, and phospho-ATGL) would provide critical mechanistic insight into the activation of these enzymes. However, our archived tissue samples have undergone partial degradation and dephosphorylation during storage, rendering them unsuitable for reliable phospho-specific protein analysis. While our current mRNA data (Figure 5) suggest altered expression of these lipogenic/lipolytic regulators, we acknowledge that phosphorylation dynamics remain unresolved in this study. We have added a section to the ​Discussion​ (Lines 467-472) to explicitly address this limitation and emphasize the need for future work to investigate post-translational modifications in Myr’s metabolic effects. Such studies will require freshly collected tissues with immediate stabilization of phosphorylation states, which we plan to prioritize in follow-up experiments. Thank you again for highlighting this important gap. Your feedback strengthens the rigor of our work and informs our future research direction.

Minor issues:

Comments 1: I In the introduction part, explanation about the association between sphingolipid metabolism and lipid/glucose regulation would be helpful.

Response 1: We thank the reviewer for this constructive suggestion. We have added sentences to the introduction to clarify the link between sphingolipid metabolism and metabolic regulation. (Lines 65-73)

Comments 2: In Table 1, adding a column showing gene function would be helpful.

Response 2: Thank you for your suggestion. While we fully agree that including gene functions would enhance clarity, doing so would introduce excessive textual detail to Table 1. To address this, we added a ‘Protein Names’ column, enabling readers to indirectly infer gene functions through established protein roles.

Reviewer 2 Report

Comments and Suggestions for Authors

The authors investigated the therapeutic potential of myriocin in counteracting dAGEs-induced obesity and its underlying mechanisms. There are, however, some concerns that need to be clarified:

  1. The authors did not upload original western blot photos as requested. When submitting original Western blot images for publication, the photos should be high-resolution and uncropped, with clear visibility of all bands, including markers.
  2. The sample numbers are not consistent. It is written under every figure that: Data were shown as mean ± SD (n = 6). Figure 1 and 2 showed 6 mice in each group, however, Figure 3 the results of serum and liver, there are only 3-4 mice in each group.
  3. It is suggested that the authors make the figures in the same format. For example, Figure 1 (D) is made as histogram with dots and SE, but Figure 1 (F) are only dots without SE.
  4. Line 301-302: Figure 3 (E) Representative images of HE of liver and adipose tissues or Oil red O staining of liver. The legend is wrong, as there are only liver images.
  5. Line 464-465: Abbreviations. The full names of some genes are not mentioned in either the Results or the Abbreviations part, for example, Hsl, Atgl, Dio2, Cidea, etc. However, the abbreviations of “AGEs, Myr, FBG, OGTT, AUC, GSP” are written twice at the end.

Reviewer 3 Report

Comments and Suggestions for Authors

The authors studied the serine palmitoyltransferase (SPT) inhibitor myriocin (Myr) and its potential use to counteract dietry induced obesity in a high-AGE diet mouse model. Overall the text is well written and very understandable.

Major comments:

The authors should discuss more their results with the existing literature in the discussion. The authors should also discuss (and/or add some more information in the Introduction) how inhibition of SPT/sphingolipid synthesis affect adipose remodelling at the molecular level.

Furthermore, as stated by the authors in the Discussion, previous publications indicated PPAR-gamma and PRDM16 to be important in adipose remodelling. Please add references for this and discuss this a bit more. To my knowledge the PPAR-gamma/PRDM16 and the PGC1alpha pathways can interact in transcriptional regulation. Do the authors imply that the AMPK-PGC1alpha pathway is the more important?

Could the authors add data on PPAR-gamma/PRDM16? Or justify why this was not done.

According to the instruction for authors of the journal, full scans of the original images must be submitted as supplementary material when cropped images are shown in figures. The authors submitted supplementary files of the cropped western blots of figures 6C and 7C. However, these supplementary files only show the cutouts used to prepare the cropped images in the figures. Images of the whole blots would be needed (together with molecular mass markers). In case the blot was cut into pieces before staining, the pieces belonging to one original blot should be reassembled for the image.

The band pattern of the blots in Figure 7C (maybe also 6C) somehow suggest that the different antibodies were not applied to the same blots, but that different blots were used for the different antibodies. If so, the authors should mention this fact in the manuscript (Figure legend) and justify it.

Minor comments:

Methods, section 2.11: Please give composition of RIPA and TBST buffers, as different recipes can be found in the literature.

line 259: use only Gck without mentioning GK

line 261: use only G6pc without mentioning G6Pase

line 323: “Overall, these indicate ...” – word missing ? “... these observations indicate ...” or “... these results indicate ...”

line 337: “Furthermore ....” instead of “And ....”

line 337: please specify: Acc1 (or Acc2 ?)

line 337: “fatty acid synthase” instead of “patty ....”

Author Response

For research article

Response to Reviewer 3 Comments

1. Summary

2. Point-by-point response to Comments and Suggestions for Authors

Comments and Suggestions for Authors

The authors studied the serine palmitoyltransferase (SPT) inhibitor myriocin (Myr) and its potential use to counteract dietry induced obesity in a high-AGE diet mouse model. Overall the text is well written and very understandable.

Major comments:

Comments 1: The authors should discuss more their results with the existing literature in the discussion. The authors should also discuss (and/or add some more information in the Introduction) how inhibition of SPT/sphingolipid synthesis affect adipose remodelling at the molecular level.

Response 1: Thank you for highlighting this. In the revised manuscript, we have added two sections to the ​Discussion​ to contextualize our findings regarding the altered expression of lipogenic/lipolytic genes and the thermogenesis-related gene UCP1 under Myr treatment, aligning these results with existing literature. We also elaborated on the potential mechanisms by which Myr may remodel adipose tissue through UCP1 upregulation, further bridging our observations to broader molecular pathways in adipose biology. (Lines 448-460, 467-471)

Comments 2: Furthermore, as stated by the authors in the Discussion, previous publications indicated PPAR-gamma and PRDM16 to be important in adipose remodelling. Please add references for this and discuss this a bit more. To my knowledge the PPAR-gamma/PRDM16 and the PGC1alpha pathways can interact in transcriptional regulation. Do the authors imply that the AMPK-PGC1alpha pathway is the more important?

Response 2: Thank you for raising this point. The publications mentioned here were provided earlier in this section. We have revised the text to clarify the referenced studies and avoid ambiguity by replacing 'prior work' with 'those studies.' As you noted, PPARγ/PRDM16 and PGC1α pathways indeed interact during transcriptional regulation in adipose remodeling, and we have expanded the discussion to include this mechanism. While our findings highlight AMPK-PGC1α signaling as a novel modulator in this context, we cannot definitively conclude whether it supersedes PPARγ/PRDM16 in importance, as comparative functional studies are lacking. We now emphasize the need for future work to dissect these interactions under Myr-induced AMPK activation.(Lines 455-459).

Comments 3: Could the authors add data on PPAR-gamma/PRDM16? Or justify why this was not done.

Response 3: Thank you for this critical insight. We agree that elucidating Myr’s effects on PPARγ/PRDM16 protein dynamics (e.g., post-translational modifications, subcellular localization, and complex formation with PGC1α) would significantly enhance the mechanistic depth of our study. While our preliminary data show Myr suppresses PPARγ mRNA level (Fig. 5), we acknowledge that protein-level analyses are essential to confirm functional relevance. Unfortunately, technical limitations currently preclude definitive conclusions. To address this gap, We plan to optimize co-IP to map PPARγ/PRDM16 interactions under Myr treatment, alongside phosphoproteomics to identify AMPK-mediated PTMs of PPARγ, PRDM16 and PGC1α. We appreciate the opportunity to clarify this limitation and have revised the Discussion to emphasize the need for future work dissecting PPARγ/PRDM16 dynamics in sphingolipid-inhibition models. (Line 458-459).

Comments 4: According to the instruction for authors of the journal, full scans of the original images must be submitted as supplementary material when cropped images are shown in figures. The authors submitted supplementary files of the cropped western blots of figures 6C and 7C. However, these supplementary files only show the cutouts used to prepare the cropped images in the figures. Images of the whole blots would be needed (together with molecular mass markers). In case the blot was cut into pieces before staining, the pieces belonging to one original blot should be reassembled for the image.

Response 4: Thank you for your feedback. We agree with your comment and have resubmitted the original Western blot images, which are now uploaded to the system.

Comments 5: The band pattern of the blots in Figure 7C (maybe also 6C) somehow suggest that the different antibodies were not applied to the same blots, but that different blots were used for the different antibodies. If so, the authors should mention this fact in the manuscript (Figure legend) and justify it.

Response 5: Thank you for highlighting this. In our Western blot analysis, certain antibody pairs (e.g., total AMPK and phospho-AMPK) exhibit cross-reactivity or require stripping protocols that risk signal degradation. Additionally, some target proteins (e.g., AMPK and Tubulin) migrate to similar regions on the gel, making sequential probing unreliable. To ensure accuracy, ​we used separate blots for distinct antibodies—a methodological detail now explicitly stated in the revised figure legends and Methods. While this approach avoids technical artifacts, we acknowledge it may introduce variability in quantification despite equal sample loading and normalization.

We sincerely appreciate your meticulous review, which also alerted us to the omission of clarifying that Figures 6C and 7C share the same loading controls (Tubulin) as they derive from identical sample sets. This detail has now been explicitly stated in the revised figure legends and Methods to enhance transparency.

Minor comments:

Comments 1: Methods, section 2.11: Please give composition of RIPA and TBST buffers, as different recipes can be found in the literature.

Response 1: Thanks for point this out. The composition of RIPA and TBST buffers are now given in the revised manuscript.

Comments 2: line 259: use only Gck without mentioning GK.

Response 2: Thanks for the advice. We now only used the word ‘Gck’.

Comments 3: line 261: use only G6pc without mentioning G6Pase.

Response 3: Thanks for the advice. We now only used the word ‘G6pc’.

Comments 4: line 323: “Overall, these indicate ...” – word missing ? “... these observations indicate ...” or “... these results indicate ...”

Response 4: Thanks for pointing this out. The word “results” was missing. We have changed this sentence into “Overall, these results indicate…”.

Comments 5: line 337: “Furthermore ....” instead of “And ....”

Response 5: Thanks for carefully reviewing. We have corrected it.

Comments 6: line 337: please specify: Acc1 (or Acc2 ?)

Response 6: It is Acc1. Thanks for point this out. We have corrected it in the revised manuscript.

Comments 7: line 337: “fatty acid synthase” instead of “patty ....”

Response 7: Thank you for catching this typographical error. We have corrected it. We sincerely apologize for the oversight and appreciate your careful reading.

Round 2

Reviewer 2 Report

Comments and Suggestions for Authors

I think the images meet the requirements, and I do not have other questions. 

Reviewer 3 Report

Comments and Suggestions for Authors

The authors have addressed all points raised by the reviewer appropriately.